# Learning Contextualized Knowledge Structures for Commonsense Reasoning

## Abstract

Recently, neural-symbolic architectures have achieved success on commonsense reasoning through effectively encoding relational structures retrieved from external knowledge graphs (KGs) and obtained state-of-the-art results in tasks such as (commonsense) question answering and natural language inference. However, these methods rely on quality and contextualized knowledge structures (*i.e.*, fact triples) that are retrieved at the pre-processing stage but overlook challenges caused by incompleteness of a KG, limited expressivity of its relations, and retrieved facts irrelevant to the reasoning context. In this paper, we present a novel neural-symbolic model, named Hybrid Graph Network (HGN), which jointly generates feature representations for new triples (as a complement to existing edges in the KG), determines the relevance of the triples to the reasoning context, and learns graph module parameters for encoding the relational information. Our model learns a compact graph structure (comprising both extracted and generated edges) through filtering edges that are unhelpful to the reasoning process. We show marked improvement on three commonsense reasoning benchmarks and demonstrate the superiority of the learned graph structures with user studies. [1]

## 1 Introduction

Commonsense knowledge is essential for developing human-level artificial intelligence systems that can understand and interact with the real world. However, commonsense knowledge is assumed by humans and thus rarely written down in text corpora for machines to learn from and make inferences with. Fig. 1 shows an example in a popular commonsense reasoning benchmark named CommonsenseQA (Talmor et al., 2019). The knowledge about the relations between concepts, *e.g.*, the fact triple (`print, Requires, use paper`), is not explicitly given in the question and answer. Without important background knowledge as clues, natural language understanding (NLU) models may fail to answer such simple commonsense questions that are trivial to humans.

Current commonsense reasoning models can be classified into retrieval-augmented methods (Banerjee et al., 2019; Pan et al., 2019) and KG-augmented methods (Wang et al., 2019b; Kapanipathi et al., 2020). Retrieval-augmented methods retrieve relevant sentences from an external corpus such as Wikipedia. The retrieved sentences are usually not interconnected, and their unstructured nature makes it inherently difficult for models to do complex reasoning over them (Zhang et al., 2018). On the other hand, symbolic commonsense KGs such as ConceptNet (Speer et al., 2017) provide structured representation of the relational knowledge between concepts, which is of critical importance for effective (multi-hop) reasoning and making interpretable predictions. Therefore, recent advances (Lin et al., 2019; Feng et al., 2020; Malaviya et al., 2020; Bosselut & Choi, 2019) have focused on KG-augmented neural-symbolic commonsense reasoning — integrating the symbolic commonsense knowledge with the pre-trained neural language models such as BERT (Devlin et al., 2019).

One of the key challenges for KG-augmented commonsense reasoning is how to obtain relevant and useful facts for the model to reason over. These supporting facts are usually not readily available to the model and require explicit annotation by humans. Most existing works (Lin et al., 2019; Wang et al., 2019b; Lv et al., 2020) follow heuristic procedures to extract supporting fact triples from KGs, *e.g.*, by finding connections between concepts mentioned in the question and answer. This simplified extraction process may be sub-optimal because the commonsense KGs are usually incomplete (Min

---

[1] Code has been uploaded and will be published.

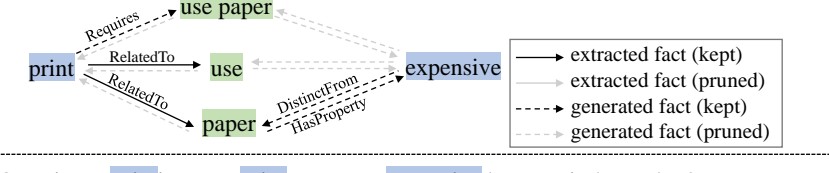

**Contextualized Knowledge Graph** for (<Question>, "use paper")

<Question>: Printing on a printer can get expensive because it does what?

<Candidate Answers>: A. explode B. use paper C. store information D. queue E. noise

Figure 1: **Commonsense question answering augmented with external graph knowledge.** Underlined words and phrases are recognized concepts. To correctly answer this question, it's desirable that the model has access to commonsense knowledge like (`print, Requires, use paper`), (`paper, HasProperty, expensive`), which is not presented in the context. This calls for the integration of contextualized commonsense knowledge.

et al., 2013) and supporting facts could be missing. To mitigate this issue, Wang et al. (2020b) fine-tune a language model to generate pseudo-paths (*i.e.*, sequences of triples) between question and answer concepts as evidence for the reasoning context (question and answer). However, when two input concepts are not closely related, the generated pseudo-paths are often unreliable as it's hard to connect two entities using a small set of predefined KG relations (*i.e.*, limited expressiveness). Besides, since KGs are context-agnostic, both extracted facts and generated facts do not necessarily relate to the central topic of the reasoning context, yielding misleading facts for reasoning. Additionally, KGs themselves store noisy facts. To summarize, low coverage of KG facts, limited expressiveness of KG relations, wrong and uncontextualized facts make neural-symbolic integration of commonsense knowledge and pre-trained language models less reliable or generalizable.

In this paper, we propose a novel KG-augmented commonsense reasoning model, named Hybrid Graph Network (HGN), to address these issues. It leverages both extracted facts (with high precision) and continuous feature representations for generated facts (with high recall) to build a *contextualized* graph with learnable edge features, which overcome the low coverage and limited expressiveness issue of the KG. It then iteratively prunes *unreliable* and *unrelated* edges during model learning, leading to a superior graph structure for reasoning. Fig. 1 shows an illustrative example of the graph structure HGN has learned. Besides triples extracted from ConceptNet, *e.g.*, (`print, RelatedTo, use`), HGN manages to (1) generate novel triples and (2) identify critical evidence triples, *e.g.*, (`print, Requires, use paper`) and (`paper, HasProperty, expensive`), while pruning triples that are unhelpful for reasoning, *e.g.*, (`use, ·, expensive`). The final contextualized graphs created by our HGN are shown to be more useful for models to reason over.

We summarize our contributions as follows: (1) We propose HGN, a KG-augmented commonsense reasoning model that overcomes the low coverage, limited expressiveness, wrong and uncontextualized facts issues of KGs. It jointly generates features for novel facts to complement extracted facts and learns the structure of the contextualized knowledge graph while reasoning over it. (2) We conduct extensive experiments on three commonsense question answering benchmarks and show consistent improvement over previous approaches. (3) We show our contextualized graph structures are more helpful for the question-answering process with user studies.

## 2 NEURAL-SYMBOLIC MODELS FOR COMMONSENSE REASONING

We focus on the task of commonsense question answering (QA), while the proposed model can be easily adapted to other tasks that require commonsense reasoning skills (*e.g.*, natural language inference). In the typical scenario of KG-augmented question answering, given a question $q$, the model is asked to select the correct answer from a set of candidate answers $\{a_i\}$ with the help of symbolic knowledge from an external knowledge graph $\mathcal{G} = \{\mathcal{E}, \mathcal{R}, \mathcal{F}\}$. Here, $\mathcal{E}, \mathcal{R}, \mathcal{F}$ denote the set of entities, relations, and facts, respectively. A fact takes the form of a triple $(h, r, t) \in \mathcal{F}$, where $h \in \mathcal{E}$ is the head entity, $t \in \mathcal{E}$ is the tail entity, and $r \in \mathcal{R}$ is their relation.

We approach the multi-choice QA problem by measuring the *plausibility* $\rho(q, a)$ between the question $q$ and **each** candidate answer $a$. The candidate answer with the highest plausibility score will be

Figure 2: **Architecture of a typical neural-symbolic model for commonsense reasoning.**

chosen as the model's prediction. Fig. 2 illustrates the workflow of a typical neural-symbolic model architecture for question answering, which our proposed model fits into. The final score is predicted based on the neural encoding of unstructured reasoning context and symbolic graph knowledge.

**Neural Encoding of Reasoning Context.** The text of the question and answer itself serves as strong unstructured evidence in evaluating their plausibility. Recent years have witnessed great success of pretrained language models (PLMs) (Devlin et al., 2019; Liu et al., 2019) in a range of NLP tasks, including question answering (Su et al., 2019; Lukovnikov et al., 2019) and natural language inference (Zhang et al., 2019; Wang et al., 2020a). Similar to previous works, here we adopt a PLM parameterized by $\theta_{\text{text}}$ to encode the question and answer pair into the statement vector: $\mathbf{s} = f_{\text{text}}([q, a]; \theta_{\text{text}})$. We use $[\cdot, \cdot]$ to denote the concatenation of sentences or vectors.

**Modeling Symbolic Graph Knowledge.** Commonsense knowledge graphs, as an external source of knowledge, can also contribute to context understanding and reasoning by providing relational knowledge between concepts related to the question and answer. For each question-candidate answer pair $(q, a)$, we build a directed graph $G = (V, E)$ with adjacentcy matrix $\mathbf{A} \in \{0, 1\}^{n \times n}$, which is termed as the *contextualized knowledge graph*. It represents the relevant knowledge structures (concepts and their relations) from the external KG. $G$'s node set $V$ includes concepts mentioned in the question and candidate answer pair $(q, a)$. Edges in $G$ represent the relations between their connected nodes. $G$ stores knowledge that is related to the context, and serves as the structured evidence for answering the question. Fig. 1 presents an example of the contextualized knowledge graph for the question and a candidate answer ("use paper").

For a contextualized KG $G$ with $n$ nodes and $m$ edges, we let $V = \{v_1, \ldots, v_n\}$. We denote the node feature vectors as $\{\mathbf{x}_i \mid v_i \in V\}$ and the edge feature vectors as $\{\mathbf{x}_{(i,j)} \mid (v_i, v_j) \in E\}$. We stack node feature vectors and edge feature vectors to get the node feature matrix $\mathbf{X} \in \mathbb{R}^{n \times d_v}$ and the edge feature matrix $\mathbf{X}_e \in \mathbb{R}^{m \times d_e}$, respectively. We denote the learnable parameters involved in the function which maps $(V, E)$ to feature embeddings $(\mathbf{X}, \mathbf{X}_e)$ as $\theta_{\text{graph-emb}}$. For encoding the contextualize knowledge graph $G$, we consider a general formulation of the graph encoder $\mathbf{g} = f_{\text{graph-enc}}(\mathbf{X}, \mathbf{X}_e, \mathbf{A}, \mathbf{s}; \theta_{\text{graph-enc}})$, parameterized by $\theta_{\text{graph-enc}}$. For simplicity, we denote it as $\mathbf{g} = f_{\text{graph}}(q, a, \mathbf{s}; \theta_{\text{graph}})$ where $\theta_{\text{graph}} = \{\theta_{\text{graph-emb}}, \theta_{\text{graph-enc}}\}$. To predict the plausibility score $\rho(q, a)$, we feed $[\mathbf{s}, \mathbf{g}]$, the concatenation of $\mathbf{s}$ and $\mathbf{g}$, to a multilayer perceptron (MLP) with parameters $\theta_{\text{MLP}}$. The output of the MLP is then passed through to a softmax layer to calculate the final probability $\hat{\rho}(q, a)$ for choosing a candidate answer, shown as follows.

$$\rho(q, a; \theta) = f_{\text{MLP}}([\mathbf{s}, \mathbf{g}]; \theta_{\text{MLP}}); \qquad \{\hat{\rho}(q, a_i; \theta)\} = \text{softmax}\{\rho(q, a_i; \theta)\}. \tag{1}$$

Here $\theta = \{\theta_{\text{text}}, \theta_{\text{graph}}, \theta_{\text{MLP}}\}$ is the set of all learnable parameters.

To derive the contextualized KG, most existing works perform heuristic graph extraction (Lin et al., 2019; Feng et al., 2020), while Wang et al. (2020b) generate relational paths to connect question and answer concepts. They all assume a perfect graph structure and fix the adjacency matrix during training. In contrast, our proposed HGN starts with an initial graph structure with both extracted and generated edges, and iteratively refines the adjacency matrix during graph encoding.

## 3 JOINTLY LEARNING GRAPH STRUCTURES AND PARAMETERS

As illustrated in Fig. 2, given a question and candidate answer $(q, a)$, we encode individual contextualized KG with a graph encoder $f_{\text{graph}}$, and then use the output graph vector $\mathbf{g}$ to estimate the plausibility of the question-answer pair. However, static graphs extracted from external KGs often suffer from limited coverage, making it hard for the model to collect and reason over adequate supporting facts. We solve the problem by considering both extracted facts and generated facts during graph initialization, resulting in a graph with "hybrid" edge features from complementary sources.

Figure 3: **Overview of our HGN's graph module. We jointly learn the graph structure and network parameters.** Darkness of edges indicate their weights. Red variables are updated in the previous step.

For the generated facts, we directly use the continuous relational features output by q generator instead of decoding into relations and re-encoding with a lookup table for better expressivity. While incorporating generated facts improve the recall, evidence precision could be impacted. Besides, the processes of extracting and generating facts are context-agnostic, leading to irrelevant facts that could mislead reasoning. Therefore, we introduce learnable edge weights to control message passing during graph encoding. We further impose entropy regularization on the learned edge weights to encourage pruning noisy edges while performing reasoning.

The overview of our graph module is shown in Fig. 3. Starting from an heuristically extracted graph (with adjacency matrix $\mathbf{A}^{\text{extract}}$), we first *generate* edge features to enrich $\mathbf{A}^{\text{extract}}$ into a graph with fully-connected edges between question and answer concepts, denoted by an adjacency matrix $\mathbf{A}^0$. Then we iteratively update the edge embeddings, the adjacency matrix and the node embeddings. The graph vector $\mathbf{g}$ is derived by pooling over node embeddings at the final layer.

Formally, we denote the label for question-answer pair $(q, a)$ as $y$, where $y = 1$ means $a$ is the correct answer to $q$ and $y = 0$ means $a$ is a wrong answer. The overall objective of jointly learning graph structure and model parameters on a training set $D_{\text{train}}$ is defined as follows:

$$\mathcal{L}(\boldsymbol{\theta}) = \sum_{(q,a,y) \sim D_{\text{train}}} \left[ \mathcal{L}_{\text{task}} \left( \hat{\rho}(q, a; \boldsymbol{\theta}) \right), y \right) + \beta \cdot \mathcal{L}_{\text{prune}}(\mathbf{A}^L(q, a; \boldsymbol{\theta}_{\text{text}}, \boldsymbol{\theta}_{\text{graph}})) \right], \qquad (2)$$

where $\boldsymbol{\theta} = \{\boldsymbol{\theta}_{\text{text}}, \boldsymbol{\theta}_{\text{graph}}, \boldsymbol{\theta}_{\text{MLP}}\}$ is the set of all learnable parameters, $\beta$ is a hyperparameter. $\mathbf{A}^L$ represents the final graph structure after $L$ layers' refinement (§3.2). $\mathcal{L}$ can be decomposed into $\mathcal{L}_{\text{task}}$ for the downstream classification task and $\mathcal{L}_{\text{prune}}$ for graph structure learning with regularization.

In the following subsections, we first introduce how we initialize the contextualized graph $G$ with densified adjacency matrix $\mathbf{A}^0$, node features $\mathbf{X}$ and hybrid edge features $\mathbf{X}_e$. Next we show how we encode the graph as $\mathbf{g} = f_{\text{graph-enc}}\left(\mathbf{X}, \mathbf{X}_e, \mathbf{A}^0, \mathbf{s}\right)$ and calculate $\mathcal{L}_{\text{task}}$ for the classification task. Finally we show how we calculate the regularization term $\mathcal{L}_{\text{prune}}$ based on the learned graph structure.

### 3.1 GRAPH INITIALIZATION

**Node Set and Node Features.** To effectively acquire knowledge from $\mathcal{G}$, we need to ground concepts in $(q, a)$ to the entity set of $\mathcal{G}$. A concept mention is defined as a text span that corresponds to an entity in $\mathcal{E}$. We perform string matching based on the concept vocabulary of ConceptNet (Speer et al., 2017) to identify concept mentions in $q$ and $a$. The sets of recognized question and answer concepts are denoted as $V^Q = \{v_i\}_{i=1}^{n_q}$ and $V^A = \{v_j\}_{j=1}^{n_a}$. For each node $v_i \in V$ where $V = V^Q \cup V^A$, we use its corresponding entity embedding $\mathbf{e}_{v_i}^{\text{ent}}$ as the node feature vector $\mathbf{x}_i$.

**Edge Feature Generator.** To account for the limited coverage issue of KGs, we build an edge feature generator $f_{\text{gen}}(\cdot, \cdot)$ that takes a subject concept and an object concept as input, and generates a vector that encodes their relation. This can be modeled as a knowledge graph completion (KGC) task. Unlike conventional KGs, commonsense KGs are much sparser (Malaviya et al., 2019), which poses challenges for standard embedding-based approaches (Yang et al., 2014; Dettmers et al., 2017). Recent works (Malaviya et al., 2019; Bosselut et al., 2019) show that pretrained language models can effectively tackle the sparsity challenge with well-learned concept semantics captured during large-scale pretraining. Also, pretrained language models themselves have proven to possess certain commonsense knowledge (Davison et al., 2019; Petroni et al., 2019). These features make pretrained language models a more ideal choice for the commonsense KG completion task. Inspired by recent success on adopting a sentence generation formulation for commonsense knowledge completion (Bosselut et al., 2019; Wang et al., 2020b), as an implementation, we define a sentence generation task where the model is asked to generate the relation tokens given the subject and the

object. Specially, each fact $(h, r, t)$ is associated with sentence in the "prompt-generation" format: $\left[\tilde{h}; \$; \tilde{t}; \$; \tilde{h}; \tilde{r}; \tilde{t}\right]$, where $\tilde{h}, \tilde{r}, \tilde{t}$ are the word sequence of $h, r, t$, $\$$ is the separator used by GPT-2. We convert all facts from ConceptNet to this format and finetune a GPT-2 on these sentences. Then we can use the finetuned GPT-2 to generate a sentence describing the relation between a given pair of subject and object. We take the hidden states during generation to compute the feature vector for better expressivity. More implementation details can be found in §A. An alternative solution is proposed in Wang et al. (2020b), where a GPT-2 is trained to generate a relational path connecting the given subject and object. The relational path has been proved to store rich information for inferring the relation (Neelakantan et al., 2015; Das et al., 2017).

**Edge Set and Hybrid Edge Features.** We build fully-connected edges between question concepts and answer concepts as they model the interactions between the question and answer, which serve as discriminative features in evaluating their plausibility. The adjacency matrix $\mathbf{A}^0$'s row-$i$ column-$j$ element is 1 if $(v_i, v_j) \in E = (V^Q \times V^A) \cup (V^A \times V^Q)$, and 0 otherwise. Note that for a large portion of $(v_i, v_j) \in E$, there may not be any fact from the knowledge graph that describes their relation. We therefore use the learned edge feature generator $f_{\text{gen}}(\cdot, \cdot)$ to generate their feature vectors. For an edge connecting concept pairs that are adjacent in $\mathcal{G}$, it can be mapped to a KG relation, and we use the relation embedding as the feature vector. Formally, for any $(v_i, v_j) \in E$, the edge feature vector $\mathbf{x}_{(i,j)}$ is calculated in a hybrid way as:

$$\mathbf{x}_{(i,j)} = \begin{cases} \mathbf{e}_r^{\text{rel}}, & \exists! r \in \mathcal{R}, s.t.(v_i, r, v_j) \in \mathcal{F}, \\ f_{\text{adapt}}(f_{\text{gen}}(v_i, v_j)), & \text{otherwise}. \end{cases} \tag{3}$$

$\mathbf{e}_r^{\text{rel}}$ is the relation embedding for KG relation $r$. $f_{\text{adapt}}(\cdot)$ is an MLP parameterized by $\theta_{\text{adapt}}$, which is used to transform the generated feature vector into the same space as $\mathbf{e}_r^{\text{rel}}$.

Since we freeze the weights of the edge feature generator $f_{\text{gen}}(\cdot, \cdot)$, learnable parameters involved in the graph initialization of our HGN is $\theta_{\text{graph-emb}} = \{\theta_{\text{ent}}, \theta_{\text{rel}}, \theta_{\text{adapt}}\}$. Here $\theta_{\text{ent}} = \{\mathbf{e}_c^{\text{ent}} \mid c \in \mathcal{E}\}$ is the set of entity embeddings and $\theta_{\text{rel}} = \{\mathbf{e}_r^{\text{rel}} \mid r \in \mathcal{R}\}$ is the set of relation embeddings.

## 3.2 GRAPH REASONING

We obtain an unweighted graph $G$ with node and edge features in §3.1. Although we use an edge feature generator to densify the connections between question and answer concepts, we are still facing the challenges of noisy edges, specifically *unreliable* edges (edges with wrong or low-quality attributes) and *unrelated* edges (edges irrelevant to answering the question), in $G$. Therefore, we propose to jointly refine the structure of the contextualized graph while performing reasoning.

**Reasoning with Weighted Graph.** To learn a graph structure for reasoning, we consider a continuous relaxation of the problem. Specifically, we generalize the unweighted graph to a weighted one and then learn to reweight all edges during reasoning. We build our graph reasoning module based on the formulation of Graph Networks (GNs) (Battaglia et al., 2018) by instantiating the layerwise node-to-edge ($v \to e$) and edge-to-node ($e \to v$) message passing functions. The edge weight indicates the helpfulness of an edge in reasoning and is used as a rescaling factor to control the message flow on this edge. Formally, the propagation rule at layer $l$ is defined as:

$$v \to e : \mathbf{h}_{(i,j)}^l = f_{v \to e}^l \left( \left[\mathbf{h}_i^{l-1}; \mathbf{h}_j^{l-1}; \mathbf{h}_{(i,j)}^{l-1}; \mathbf{s}\right] \right); w_{(i,j)}^l = f_w^l \left( \left[\mathbf{h}_{(i,j)}^l; \mathbf{s}\right] \right); \mathbf{A}_{(i,j)}^l = \frac{e^{w_{(i,j)}^l}}{\sum_{(s,t) \in E} e^{w_{(s,t)}^l}},$$

$$e \to v : \mathbf{u}_{(i,j)}^l = f_u^l \left( \left[\mathbf{h}_i^{l-1}; \mathbf{h}_{(i,j)}^l\right] \right); \mathbf{h}_j^l = f_{e \to v}^l \left( \sum_{i \in \mathcal{N}_j} \mathbf{A}_{(i,j)}^l \mathbf{u}_{(i,j)}^l \right).$$

$$\tag{4}$$

$N_j$ is the set of $v_j$'s incoming neighbors, $f_{v \to e}^l, f_w^l, f_u^l$ and $f_{e \to v}^l$ are MLPs, $\mathbf{h}_{(i,j)}^0 = \mathbf{x}_{(i,j)}, \mathbf{h}_i^0 = \mathbf{x}_i$.

In node-to-edge message passing, for each edge $(v_i, v_j) \in E$, we calculate its updated edge embedding $\mathbf{h}_{(i,j)}^l$ which encodes the corresponding fact. We also assign to it an unnormalized score $w_{(i,j)}^l$ which measures both the validness of the fact itself (captured by $\mathbf{h}_{(i,j)}^{l-1}$) and how helpful it is to the reasoning context (captured by $\mathbf{s}$). The edge score is globally normalized across all edges to become the edge weight $\mathbf{A}_{(i,j)}^l$, so that edges with low scores will be softly pruned by receiving a close-to-zero weight. We choose global normalization instead of local normalization (normalization within

each node's neighborhood) like Graph Attention Network (GAT) (Veličković et al., 2017) because local normalization assumes at least one edge should be helpful in a node's incoming neighborhood, which is not true in our situation. For example, for the distracting or wrongly-grounded concepts, none of their connected edges could be helpful and all edges in the neighborhood should be pruned. Those noisy nodes should be softly excluded from message passing rather than still receive message from a weighted combination of their neighbors. We also empirically compare our proposed model with a variant using GAT-like edge attention in §4.3.

In edge-to-node message passing, we calculate the message vector $\mathbf{u}^l_{(i,j)}$ based on the start node and edge embeddings. We use the edge weight to rescale the message vector and each node aggregates messages from incoming neighbors to update its node embedding.

Once we get the final node embeddings after $L$ layers' message passing, we aggregate them into a graph-level representation through a graph pooling operation. As different nodes can have distinct informativeness, we employ an attention mechanism, where nodes are assigned different importance, to obtain the graph encoding $\mathbf{g}$: $\alpha_i = \mathbf{s}\mathbf{W}_{\text{att}}\mathbf{h}^L_i, \mathbf{g} = \sum_{i:v_i \in V} \frac{e^{\alpha_i}}{\sum_{j:v_j \in V} e^{\alpha_j}} \mathbf{h}^L_i$. Here $\mathbf{W}_{\text{att}}$ is a learnable matrix for calculating each attention score $\alpha_i$ for node $v_i$. Given $\mathbf{g}$, we calculate $\hat{\rho}(q, a)$ using Eq. 1. We adopt the cross-entropy loss for the main classification task:

$$\mathcal{L}_{\text{task}}(\hat{\rho}(q, a; \boldsymbol{\theta})), y) = -y \log \hat{\rho}(q, a; \boldsymbol{\theta}). \tag{5}$$

**Learning to Prune Edges with Entropy Regularization.** To encourage the model to take decisive pruning steps on the graph structure, we add a regularization term to the loss function to penalize non-discriminative edge weights. In an extreme case, a blind model will assign the same weight to all edges and $G$ is degenerated into an unweighted graph, where usually a big number of noisy edges are mixed with a small number of helpful edges. Therefore, to guide the model to discriminate the helpful edges in the reasoning process, we minimize the entropy of the edge weight distribution as an auxiliary training objective. The motivation is that the information entropy can be used to measure the informativeness of the edge weight predictions. A lower entropy, caused by a skewed distribution, means the model is actively incorporating more priors (*e.g.* the plausibility of its corresponding fact and the relatedness to the question-answer pair) into edge weight prediction. Formally, the entropy regularization term of $G$ is calculated as:

$$\mathcal{L}_{\text{prune}}(\mathbf{A}^L(q, a; \boldsymbol{\theta}_{\text{text}}, \boldsymbol{\theta}_{\text{graph}})) = - \sum_{(i,j):(v_i, v_j) \in E} \mathbf{A}^L_{(i,j)} \log \mathbf{A}^L_{(i,j)}. \tag{6}$$

We add the penalty term to the downstream classification loss so that the graph structure can be jointly learned with graph reasoning. We train our model end-to-end by minimizing the overall objective function $\mathcal{L}(\boldsymbol{\theta})$ in Eq. 2 using RAdam (Liu et al., 2020) optimizer.

## 4 EXPERIMENTS

### 4.1 EXPERIMENT SETUP

We evaluate our proposed model on three multiple-choice commonsense QA datasets: **CommonsenseQA** (Talmor et al., 2019), **OpenbookQA** (Mihaylov et al., 2018) and **CODAH** (Chen et al., 2019) (details in §B). We use ConceptNet (Speer et al., 2017), a commonsensense knowledge graph, as $\mathcal{G}$. For text encoder $f_{\text{text}}$, we experiment with BERT-base, BERT-large (Devlin et al., 2019) and RoBERTa (-large) (Liu et al., 2019) to validate our model's effectiveness over different text encoders. For OpenbookQA, retrieving related facts from the openbook plays an important role in boosting the model's performance. Therefore, we also build our graph reasoning model on top of a retrieval-augmented method "AristoRoBERTa". In this way, we can study if strong retrieval-augmented methods could still benefit from KG knowledge and our reasoning framework.

### 4.2 COMPARED METHODS

We compare our model with a series of KG-augmented methods:

**Models Using Extracted Facts: RN** (Santoro et al., 2017) builds the graph with the same node set as our method but extracted edges only. The graph vector is calculated as $g = \text{Pool}(\{\text{MLP}([\mathbf{x}_i; \mathbf{x}_{(i,j)}; \mathbf{x}_j]) \mid (v_i, r, v_j) \in \mathcal{F}\})$. **GN** (Battaglia et al., 2018) presents a general formulation of GNNs. We instantiate it with the layerwise propagation rule defined in Eq. 4. It

| Methods | BERT-Base | | BERT-Large | | RoBERTa | |
|---|---|---|---|---|---|---|
| | 60% Train | 100% Train | 60% Train | 100% Train | 60% Train | 100% Train |
| LM Finetuning | 52.06 ($\pm$0.72) | 53.47 ($\pm$0.87) | 52.30 ($\pm$0.16) | 55.39 ($\pm$0.40) | 65.56 ($\pm$0.76) | 68.69 ($\pm$0.56) |
| RN (Santoro et al., 2017) | 54.43 ($\pm$0.10) | 56.20 ($\pm$0.45) | 54.23 ($\pm$0.28) | 58.46 ($\pm$0.71) | 66.16 ($\pm$0.28) | 70.08 ($\pm$0.21) |
| RN + Link Prediction | - | - | 53.96 ($\pm$0.56) | 56.02 ($\pm$0.55) | 66.29( $\pm$0.29) | 69.33 ($\pm$0.98) |
| RGCN (Schlichtkrull et al., 2018b) | 52.20 ($\pm$0.31) | 54.50 ($\pm$0.56) | 54.71 ($\pm$0.37) | 57.13 ($\pm$0.36) | 68.33 ($\pm$0.85) | 68.41 ($\pm$0.66) |
| GN (Battaglia et al., 2018) | 53.67 ($\pm$0.45) | 55.65 ($\pm$0.51) | 54.78 ($\pm$0.61) | 57.81 ($\pm$0.67) | 68.78 ($\pm$0.67) | 71.12 ($\pm$0.45) |
| GconAttn (Wang et al., 2019a) | 51.36 ($\pm$0.98) | 54.41 ($\pm$0.50) | 54.96 ($\pm$0.69) | 56.94 ($\pm$0.77) | 68.09 ($\pm$0.63) | 69.88 ($\pm$0.47) |
| KagNet (Lin et al., 2019) | - | 56.19 | | 57.16 | - | - |
| MHGRN (Feng et al., 2020) | 54.12 ($\pm$0.49) | 56.23 ($\pm$0.82) | 56.76 ($\pm$0.21) | 59.85 ($\pm$0.56) | 68.84 ($\pm$1.06) | 71.11 ($\pm$0.81) |
| PathGenerator (Wang et al., 2020b) | - | - | 55.47 ($\pm$0.92) | 57.21 ($\pm$0.45) | 68.65 ($\pm$0.02) | 71.55 ($\pm$0.99) |
| HGN (w/o edge weights) | 54.45 ($\pm$0.39) | 56.56 ($\pm$0.67) | 55.67 ($\pm$0.65) | 58.89 ($\pm$0.45) | 70.01 ($\pm$0.91) | 72.09 ($\pm$0.87) |
| HGN | **55.39** ($\pm$0.34) | **57.82** ($\pm$0.23) | **57.23** ($\pm$0.56) | **60.43** ($\pm$0.54) | **70.34** ($\pm$0.79) | **72.88** ($\pm$0.83) |

Table 1: **Accuracy on CommonsenseQA inhouse test set.** We use the inhouse split as Lin et al. (2019). Some of the baseline results are reported by Feng et al. (2020) and Wang et al. (2020b). Mean and standard deviation of four runs are presented for all models except KagNet.

| Methods | RoBERTa | AristoRoBERTa |
|---|---|---|
| LM Finetuning | 64.80 ($\pm$2.37) | 77.40 ($\pm$1.64) |
| RN (Santoro et al., 2017) | 63.65 ($\pm$2.31) | 75.35 ($\pm$1.39) |
| RN + Link Prediction | 66.30 ($\pm$0.48) | 77.25 ($\pm$1.11) |
| RGCN (Schlichtkrull et al., 2018b) | 62.45 ($\pm$1.57) | 74.60 ($\pm$2.53) |
| GN (Battaglia et al., 2018) | 66.20 ($\pm$2.14) | 77.25 ($\pm$0.91) |
| GconAttn (Wang et al., 2019a) | 64.75 ($\pm$1.48) | 71.80 ($\pm$1.21) |
| MHGRN (Feng et al., 2020) | 66.85 ($\pm$1.19) | 77.75 ($\pm$0.38) |
| PathGenerator (Wang et al., 2020b) | 68.40 ($\pm$0.31) | **80.05** ($\pm$0.68) |
| HGN (w/o edge weights) | 67.20 ($\pm$2.07) | 78.45 ($\pm$0.46) |
| HGN | **69.00** ($\pm$0.95) | 79.00 ($\pm$1.43) |

Table 2: **Test accuracy on OpenbookQA.** Some of the baseline results are reported by Feng et al. (2020) and Wang et al. (2020b). Mean and standard deviation of four runs are presented for all models.

| Methods | Text Encoder | Test Acc |
|---|---|---|
| UnifiedQA (Khashabi et al., 2020) | T5 | 87.2 |
| T5 + KB | T5 | 85.4 |
| T5 (Raffel et al., 2020) | T5 | 83.2 |
| PathGenerator (Wang et al., 2020b) | AristoAlbert | 81.8 |
| **HGN (ours)** | **AristoRoBERTa** | **81.4** |
| AristoRoBERTa + KB | AristoRoBERTa | 81.0 |
| MHGRN (Feng et al., 2020) | AristoRoBERTa | 80.6 |
| PathGenerator (Wang et al., 2020b) | AristoRoBERTa | 80.2 |
| KF + SIR (Banerjee & Baral, 2020) | RoBERTa | 80.2 |
| AristoRoBERTa | AristoRoBERTa | 80.2 |

Table 3: **Leaderboard of OpenbookQA.** Our HGN ranks first among all submissions using AristoRoBERTa as the text encoder.

differs from our HGN in that: (1) it only considers extracted edges; (2) all edge weights are fixed to 1. **MHGRN** (Feng et al., 2020) generalizes GNNs with multi-hop message passing. Descriptions for **RGCN**, **GconAttn**, and **KagNet** can be found in §D.

**Models Using Generated Facts: RN + Link Prediction** differs from RN by only considering the generated relation (predicted using TransE (Bordes et al., 2013)) between question and answer concepts. **PathGenerator**[23] (Wang et al., 2020b) learns a path generator from paths collected through random walks on the KG. The learned generator is used to generate paths connecting question and answer concepts. Attentive pooling is used to derive the graph vector given a set of path embeddings.

**Our Model's Variant: HGN (w/o edge weights)** reasons over an unweighted graph with hybrid features, which means edge weights are fixed to 1 during training.

### 4.3 RESULTS

**Performance Comparisons.** Tables 1, 2 and 4 show performance comparisons between our models and baseline models on CommonsenseQA, OpenbookQA, and CODAH respectively. Our HGN shows consistent improvement over baseline models on all datasets except that it achieves the second best performance on OpenbookQA with AristoRoBERTa as the text encoder. We also submit our best model to OpenbookQA's leaderboard.[4] Our model ranks the first among all models using AristoRoBERTa as the text encoder,[5] demonstrating the effectiveness of our proposed model. As a comparison, most baseline models fail to achieve further improvement over it. That may be because AristoRoBERTa has access to knowledge collected through retrieval, which makes it difficult to further benefit from KG knowledge if a weak reasoning approach is adopted. The improvement

---

[2]PathGenerator is a contemporaneous work that learns a edge feature generator based on multi-hop paths, which has greater expressive power than our (1-hop) edge feature generator. Our reasoning framework is compatible with any implementation for $f_{\text{gen}}$, and we will build on PathGenerator in our future experiments.

[3]We choose PG-Global as the representative variant for PathGenerator as it performs better than PG-Local. PG-Full is an ensemble model of PG-Global and RN, so we don't consider it in our comparisons.

[4]We tried more than 4 seeds for the leaderboard submission, which is different from the setting of Table 2.

[5]T5 has 11B parameters, making it impractical for us to finetune with available computational resources.

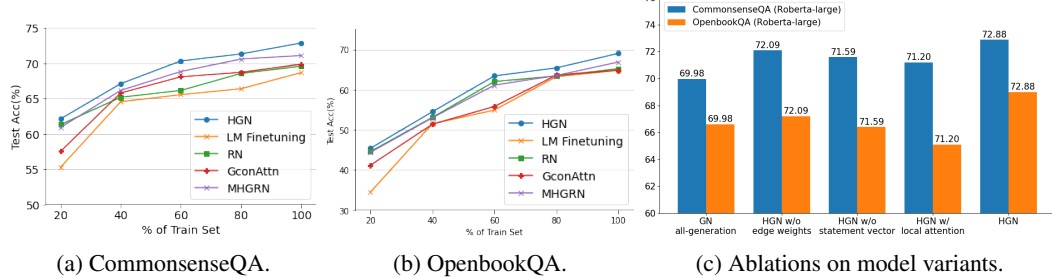

(a) CommonsenseQA.  (b) OpenbookQA.  (c) Ablations on model variants.

Figure 4: **Ablation studies.** (a)(b) Performance of HGN and baseline models with different amount of training data; (c) Performance of different model variants.

| Methods | BERT-large | RoBERTa |
|---|---|---|
| LM Finetuning | 65.89 | 83.20 |
| RN (Santoro et al., 2017) | 66.24 | 82.85 |
| RGCN (Schlichtkrull et al., 2018b) | 65.27 | 82.24 |
| GN (Battaglia et al., 2018) | 65.89 | 82.24 |
| MHGRN (Feng et al., 2020) | 66.17 | 83.21 |
| HGN (w/o edge weights) | 66.28 | 83.25 |
| HGN | **66.71** | **83.75** |

Table 4: **Test accuracy on CODAH.** We use the same 5-fold cross validation as Yang et al. (2020).

| Contextualized Graph | GN ($\mathbf{A}^{\text{extract}}$) | HGN ($\mathbf{A}^K$) |
|---|---|---|
| Number of Edges | 3.65 (±2.73) | 4.38 (±3.24) |
| Number of Valid Edges | 2.67 (±1.95) | 3.15 (±1.98) |
| Percentage of Valid Edges | 71.64% | 78.51% |
| Average Helpfulness Score of Edges | 0.90 (±0.50) | 1.16 (±0.51) |
| Prune Rate | - | 77.13% |

Table 5: **User studies on learned graph structures.** 30 pairs of contextualized graphs output by GN and HGN are evaluated by 5 annotators.

on CODAH is less significant compared to the other two datasets. As Chen et al. (2019) suggest, questions in CODAH mainly target commonsense reasoning about quantitative, negation and object reference. In this case, relational knowledge provided by ConceptNet may only offer limited help.

From the results of GN and our model variants, we can see "HGN (w/o edge weights)" consistently outperforms GN, which means generated facts could help reasoning, and suggests that current KG-augmented models can be improved with a more complete KG. Our HGN further improves over "HGN (w/o edge weights)", indicating the effectiveness of conducting context-dependent pruning.

**Training with Less Labeled Data.** Fig. 4 (a)(b) show the results of our model and baseline models when trained with 20%/40%/60%/80%/100% of the training data on CommonsenseQA and OpenbookQA. Our model gets better test accuracy under all settings. The improvement over the knowledge-agnostic baseline (LM Finetuning) is more significant with less training data, which suggests that incorporating external knowledge is more helpful in the low-resource setting.

**Study on More Model Variants.** To better understand the model design, we experiment with two more variants on CommonsenseQA and OpenbookQA. **GN all-generation** doesn't consider extracted facts and instead generate edge features between all question and answer concepts. **HGN w/o statement vector** doesn't consider s in Eq. 4, which isolates the graph encoder from the text encoder. **HGN w/ local attention** replaces our edge weight design with a local attention mechanism similar to GAT (details in §D). Fig. 4 (c) shows the results of the ablation study. Comparing "GN all-generation" with "HGN w/o edge weights", we can conclude that extracted facts play an important role in HGN and can't be replaced by generated features. The high precision of extracted facts is still desirable even if we have a model to generate relational edges. Comparing "HGN w/o statement vector" with "HGN", we find that accessing context information is also important for graph reasoning, which means information propagation and edge weight prediction should be conducted in a context-aware manner. "HGN w/ local attention" is outperformed by HGN, suggesting that "global edge weight" is a more appropriate design choice than "local edge attention".

### 4.4 USER STUDIES ON LEARNED GRAPH STRUCTURES

To assess our model's ability to refine the graph structure, we compare the graph structure before and after being processed by HGN. Specifically, we sample 30 questions with its correct answer from the development set of CommonsenseQA and ask 5 human annotators to evaluate the graph output by GN (with adjacency matrix $\mathbf{A}^{\text{extract}}$ and extracted facts only) and our HGN (with adjacency matrix $\mathbf{A}^L$). We manually binarizing $\mathbf{A}^L$ by removing edges with weight less than 0.01.

Given a graph, for each edge (fact), annotators are asked to rate its **validness** and **helpfulness**. The validness score is rated as a binary value in a context-agnostic way: 0 (the fact doesn't make sense), 1 (the fact is generally true). The helpfulness score measures if the fact is helpful for solving the question and is rated on a 0 to 2 scale: 0 (the fact is unrelated to the question and answer), 1 (the fact is related but doesn't directly lead to the answer), 2 (the fact directly leads to the answer). The mean ratings for 30 pairs of (GN, HGN) graphs by 5 annotators are reported in Table 5. We also include another metric named "prune rate" calculated as: $1 - \frac{\# \text{ edges in binarized } \mathbf{A}^K}{\# \text{ edges in } \mathbf{A}^0}$, which measures the portion of edges that are assigned very low weights (softly pruned) during training and is only applicable to HGN. The Fleiss' Kappa (Fleiss, 1971) is 0.51 (moderate agreement) for validness and 0.36 (fair agreement) for helpfulness. The graph refined by HGN has both more edges and more valid edges compared to the extracted one. The refined graph also achieves a higher helpfulness score. These all indicate that our HGN learns a superior graph structure with more helpful edges and less noisy edges, which is the reason for performance improvement over previous works that rely on extracted and static graphs. Detailed cases can be found in §C.

## 5 RELATED WORK

**Commonsense Question Answering.** Answering commonsense questions is challenging because the required commonsense knowledge is neither written down in the context nor held by pretrained language models. Therefore, many works leverage external knowledge to obtain additional evidence. These works can be categorized into IR-augmented methods, where evidence is retrieved from text corpora, and KG-augmented methods, where evidence is collected from KGs. Lv et al. (2020) demonstrate that IR-based evidence and KG-based evidence are complementary to each other. Although adding IR evidence can lead to further performance improvement, we only focus on the challenges of KG-augmented methods in this paper. Literature in this domain mainly studies how to encode the contextualized subgraph extracted from a KG. For example, Lin et al. (2019) propose a model comprised of GCN and LSTM to account for both the global graph structure and local paths connecting question concepts and answer concepts. Ma et al. (2019) use BERT to generate the embedding for the pseudo-sentence representing each edge and then adopt the attention mechanism to aggregate edge features as the graph encoding. Crucial difference is that they assume a static graph and there's no operation on enriching or denoising the graph structure. While Wang et al. (2020b) also complete the contextualized graph with a path generator, they still reason over the static graph and neglect the noise introduced during generation.

**Graph Structure Learning.** Works that jointly learn the graph structure with the downstream task can be classified into two categories. One line of works directly learn an unweighted graph with desired edges for reasoning. Kipf et al. (2018) and Franceschi et al. (2019) sample the graph structure from a predicted probabilistic distribution with differentiable approximations. Norcliffe-Brown et al. (2018) calculate the relatedness between any pair of nodes and only keep the top-$k$ strongest connections for each node to construct the edge set. Sun et al. (2019) start with a small graph and iteratively expand it with a set of retrieving operations. The other line of works consider a weighted graph with all possible edges and softly filter out the noisy ones by downweighting them. An adjacency matrix with continuous values is incorporated into message passing. Jiang et al. (2019) and Yu et al. (2019) use heuristics to regularize the learned adjacency matrix. Hu et al. (2019) consider the question embedding for predicting edge weights. Our HGN falls into the second category and therefore avoids information loss caused by hard pruning and approximation. Our uniqueness is that we construct the graph with hybrid features based on extracted and generated facts and we let node features, edge features, edge weights, and the global signal (statement vector) collectively determine the evolution of the graph structure. These empower our model with greater capacity and flexibility for KG-augmented QA.

## 6 CONCLUSION

In this paper, we propose a neural-symbolic framework for commonsense reasoning named HGN. To address the issues with missing facts from external knowledge graph and noisy facts from the contextualized knowledge graph, our proposed HGN jointly generates features for new edges, refines the graph structure, and learns the parameters for graph networks. Experimental results and user studies demonstrate the effectiveness of our model. In the future, we plan to incorporate open relations in graph initialization, which are more expressive than predefined KG relations. We also plan to study how to make the fact generation or extraction process aware of the reasoning context.

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

## A   IMPLEMENTATION DETAILS OF EDGE FEATURE GENERATOR

As an implementation of $f_{\text{gen}}$, we adopt GPT-2 (Radford et al., 2019), which is pretrained on large corpora and achieves great success on a wide range of tasks involving sentence generation, as a generator to generalize the facts from the knowledge graph. We first convert each fact $(h, r, t) \in \mathcal{F}$ into a word sequence with a "prompt-generation" format: $\left[\tilde{h}, \$, \tilde{t}, \$, \tilde{h}, \tilde{r}, \tilde{t}\right]$, where $\tilde{h}, \tilde{r}, \tilde{t}$ are the word sequence of $h, r, t$ respectively, $\$$ denotes the delimiter token used by GPT-2, and $[\cdot; \cdot]$ denotes word sequence concatenation. We denote the synthetic sentence as $s_{(h,r,t)} = \left[x_1^{(h,r,t)}, \ldots, x_{n_{(h,r,t)}}^{(h,r,t)}\right]$ and finetune GPT-2 on all synthetic sentences created from $\mathcal{F}$ with the language modeling objective:

$$\mathcal{L}_{\text{gen}}(\mathcal{F}) = \sum_{(h,r,t)\in\mathcal{F}} \sum_{i=1}^{n_{(h,r,t)}} \log P\left(x_i^{(h,r,t)} \mid x_1^{(h,r,t)}, \ldots, x_{i-1}^{(h,r,t)}\right). \tag{7}$$

After that, given any two concepts $(v_i, v_j)$, we build a prompt as $[\tilde{v}_i; \$; \tilde{v}_j; \$]$ and let the model to generate the following word sequence. We denote the whole sentence (both prompt and generation) as $s_{(v_i, v_j)}$, and the hidden states of each word during generation as $\mathbf{h}_1, \ldots, \mathbf{h}_T$ where $T$ is the sentence length. We average hidden states of all words in the sentence to get the relational feature: $f_{\text{gen}}(v_i, v_j) = \frac{1}{T} \sum_{i=1}^{T} \mathbf{h}_i$.

## B   DETAILS OF DATASETS

**CommonsenseQA** (Talmor et al., 2019) is a multiple-choice QA dataset targeting commonsense. It's constructed based on the knowledge in ConceptNet. Since the test set of the official split (9741/1221/1140 for OFtrain/OFdev/OFtest) is not publicly available, we compare our models with baseline models on the inhouse split (8500/1221/1241 for IHtrain/IHdev/IHtest)[6] used by previous works (Lin et al., 2019; Feng et al., 2020; Wang et al., 2020b).

**OpenbookQA** (Mihaylov et al., 2018) is a multiple-choice QA dataset modeled after openbook exams. Besides 5957 elementary-level science questions (4957/500/500 for train/dev/test), it also provides an open book with 1326 core science facts. Solving the dataset requires combining facts from open book with commonsense knowledge.

**CODAH** (Chen et al., 2019) contains 2801 sentence completion questions testing commonsense reasoning skills. We perform 5-fold cross validation following the practice in Yang et al. (2020). The authors shared their splits with us.

## C   CASE STUDY

We compare the graph generated by our HGN with the extracted one (GN). On the development set, there are two dominating cases and we show the representative instance of each one. Figure 5 shows the first case, where HGN prunes edges from the extracted graph. Our HGN assigns the highest weights to the most helpful facts (`book`, `AtLocation`, `house`), (`telephone book`, `AtLocation`, `house`). It also downweight unhelpful fact (`place`, `IsA`, `house`) and invalid fact (`usually`, `RelatedTo`, `house`). Figure 6 shows the second case, where new generated facts are incorporated into reasoning. All generated facts that are kept by the model make sense in the context and help identify the answer. Both cases suggest that our model improve the quality of the contextualized knowledge graph compared to the current methods that only rely on extracted facts.

## D   COMPARED METHODS

**RGCN** (Schlichtkrull et al., 2018a) extends Graph Convolutional Networks (GCNs) (Kipf & Welling, 2017) with relation-specific transition matrices during message passing. It operates on the same graph as RN. The graph vector is calculated as $\mathbf{g} = \text{Pool}(\{\mathbf{h}_i^K \mid v_i \in V\})$.

**GconAttn** (Wang et al., 2019b) softly aligns the nodes in question and answer and do pooling over all matching nodes to get $\mathbf{g}$.

---

[6]https://github.com/INK-USC/MHGRN/blob/master/data/csqa/inhouse_split_qids.txt

**Question**: What is a place that usually does not have an elevator and that sometimes has a telephone book?

**Answer**: house

**Triples**:

(book, AtLocation, house), **Edge weight:** 0.48, **Edge type:** extracted — Graph of HGN

(telephone book, AtLocation, house), **Edge weight:** 0.48, **Edge type:** extracted

(place, IsA, house), **Edge weight:** 0.01, **Edge type:** extracted — Graph of GN

(usually, RelatedTo, house), **Edge weight:** 0.01, **Edge type:** extracted

Figure 5: **Case I: Unrelated extracted facts are filtered out.**

**Question**: Where would you find an office worker gossiping with their colleagues?

**Answer**: water cooler

**Triples**:

(gossip, RelatedTo, water cooler), **Edge weight:** 0.09, **Edge type:** extracted — Graph of GN

(office, RelatedTo, cooler), **Edge weight:** 0.09, **Edge type:** extracted

(office, RelatedTo, water), **Edge weight:** 0.09, **Edge type:** extracted

(office, RelatedTo, water cooler), **Edge weight:** 0.09, **Edge type:** extracted

(office worker, AtLocation, water cooler), **Edge weight:** 0.02, **Edge type:** generated

(worker, AtLocation, water cooler), **Edge weight:** 0.02, **Edge type:** generated

(gossiping, AtLocation, water cooler), **Edge weight:** 0.02, **Edge type:** generated — Graph of HGN

Figure 6: **Case II: Helpful generated facts are incorporated.**

**KagNet** (Lin et al., 2019) uses an LSTM to encode relational paths between question and answer concepts and pool over the path embeddings for graph encoding.

**HGN w/ local attention** differs from HGN in that the edge weight is normalized in each node's neighborhood instead of the whole graph. The entropy regularization is not applicable for this variant.

