# OpenReview forum: "Learning Contextualized Knowledge Structures for Commonsense Reasoning"
_ICLR.cc/2021/Conference — Reject_

### Official Review · AnonReviewer1 · 2020-10-28
**Well evaluated and effective KG-based commonsense QA framework**

**Rating:** 7
**Confidence:** 3

**Review:**

=== Summary ===

In this paper, the authors propose a new approach towards incorporating knowledge graphs (KG) into commonsense QA frameworks. KGs are helpful for adding structured "world" information, which neural-symbolic architectures can leverage to do commonsense reasoning, e.g., "what is the expensive resource in printing on paper?" (paper). In such architectures, however, the authors argue that KG quality is a large impediment (e.g., missing or incorrect edges, distracting nodes, etc). Therefore, they propose a "hybrid" KG-based model (accordingly named "Hybrid Graph Network") that jointly learns to refine/augment the graph structure while also optimizing it for inference performance.

Experiments are conducted on a number of commonsense reasoning tasks with multiple KG sources, and compared to relevant baselines. They also perform a user study to examine the "helpfulness" of the refined KGs produced by the HGN.

=== Justification for Score ===

This paper is well-written and well-evaluated. The proposed method is also relatively simple and intuitively motivated. The experiments, however, only show modest (yet still positive) empirical gains. Perhaps not a game-changer for commonsense QA, but still a reasonable contribution that I would recommend for acceptance.

=== Strengths ===

+ The paper is clear and well-written.
+ The experimental section is strong. The model is compared to strong baselines, and I appreciated the extra user-study on learned graph structure.
+ The method is well-motivated, and provides (modest) empirical gains compared to some baselines.
+ The method shows good performance with respect to increasing data efficiency (Fig. 4).

=== Concerns ===

- The main concern is on the empirical effectiveness of the model. The results appear to give only modest gains at best (against comparable baselines to the best of my knowledge). For a number of the results the variance is large compared to the relative difference---it would helpful to also include tests of significance for these improvements.

- On OpenbookQA the model significantly underperforms T5-based models. Though I appreciate T5 is unwieldy due to its large size, it makes me question if this method indeed presents a complimentary gain, or is climbing the wrong architectural hill.

=== Update After Rebuttal ===

I commend the authors on a through rebuttal and active rewrites/experimentation. I still think the work is good, and can warrant acceptance. However, I still find the empirical results to be only moderate at best (though I appreciated the authors' rebuttal and significance testing). I am keeping my score the same.

---

> ### Author Response · Authors · 2020-11-20
> **Response to Reviewer 1**
>
> Thanks for your valuable feedback!
>
> ### Concerns 1
> >The main concern is on the empirical effectiveness of the model. The results appear to give only modest gains at best (against comparable baselines to the best of my knowledge). For a number of the results the variance is large compared to the relative difference---it would helpful to also include tests of significance for these improvements.
>
> Thanks for pointing this out. You can refer to our general response for the discussion on the empirically performance gains, significance of improvements, and other merits of our proposed model. We equip our HGN with a stronger edge feature generator. Without hyperparameter tuning, it significantly outperforms all baselines on CommonsenseQA and all baselines except PathGenerator on OpenbookQA with p-value < 0.05.
>
> For our original HGN:
> - on 60% CommonsenseQA + BERT-Base: significant over all baselines
> - on 100% CommonsenseQA + BERT-Base: significant over all baselines
> - on 60% CommonsenseQA + BERT-Large: significant over all baselines except MHGRN
> - on 100% CommonsenseQA + BERT-Large: significant over all baselines except MHGRN
> - on 60% CommonsenseQA + RoBERTa: significant over all baselines except MHGRN
> - on 100% CommonsenseQA + RoBERTa: significant over all baselines except PathGenerator
> - on OpenbookQA + RoBERTa: significant over all baselines except PathGenerator
> - on OpenbookQA + AristoRoBERTa: significant over RN, RGCN, GconAttn
> - on CODAH: significant over RGCN on RoBERTa
>
> Note that MHGRN and PathGenerator are both to be published in EMNLP 2020. For CODAH, we analyze in §4.3 “Performance Comparisons” that: “As Chen et al. (2019) suggest, questions in CODAH mainly target commonsense reasoning about quantitative, negation and object reference. In this case, relational knowledge provided by ConceptNet may only offer limited help.”
>
> We’ll do more exploration and tuning on our new model to include stronger results.
>
> ### Concerns 2
> >On OpenbookQA the model significantly underperforms T5-based models. Though I appreciate T5 is unwieldy due to its large size, it makes me question if this method indeed presents a complimentary gain, or is climbing the wrong architectural hill.
>
> T5 is a stronger pretrained model compared to BERT, RoBERTa, etc. and established new SOTA results on a series of benchmarks. However, full T5 has 11B parameters, which is 30 times larger than RoBERTa-large (355M parameters). The huge size makes it impractical to be widely studied and deployed due to lack of computing infrastructure.
>
> We believe in the potential of our model even with T5 as decoder based on two observations:
> 1. On ComonsenseQA and OpenbookQA, with BERT-Base, BERT-Large and RoBERTa being the text encoders, our model consistently outperforms knowledge-augmented baselines, which are generally better than the knowledge-agnostic baseline (LM Finetuning). It suggests that, if a knowledge-agnostic model can be improved with existing knowledge-augmented reasoning methods, then it can get further enhancement with our proposed HGN which resolves the coverage and quality issues of the supporting knowledge facts that are of vital importance in knowledge-augmented reasoning.
> 2. Although T5 hasn't been widely tested on many benchmarks, results on OpenbookQA's leaderboard suggest that T5 can also benefit from KG knowledge (T5 got 83.2 test acc while T5+KB got 85.4 test acc, as shown in Table 3 in the paper).
>
> Therefore, we believe incorporating external (KG) knowledge is still an important direction towards advanced commonsense reasoning abilities. The key challenge is how to effectively collect and reason over high-quality knowledge facts, which our model is targeting at.

---

> ### Author Response · Authors · 2020-11-24
> **Look Forward to Hearing from You**
>
> Dear Reviewer 1,
>
> We really appreciate your positive feedback and thoughtful comments! We have responded to your concerns with more discussion on empirical effectiveness and other advantages of our model, as well as the value of our proposed method over huge pretrained models like T5. We're wondering if our response addresses your concern? We really appreciate it if you could let us know any further comments before the end of the rebuttal period.
>
> Thanks again for your time!

---

### Official Review · AnonReviewer3 · 2020-10-28

**Rating:** 6
**Confidence:** 4

**Review:**

The paper proposes a question answering model that is augmented with a common-sense knowledge graph (KG). The paper builds on the following two observations — (a) KGs are incomplete often lacking facts that would be needed for reasoning to answer a question. (b) Current methods over-retrieves facts (edges) from the KG leading to a lot of unrelated facts that potentially makes reasoning noisier and harder.

The paper first retrieves all possible facts from the KG connecting entities in the question and answer. However, due to the incompleteness of the KG, the retrieved subgraph might be missing important edges between entities. To deal with this, they connect all nodes between question and answer entities and initialize the embedding of the newly added edge with hidden layers of a sentence generated from a fine-tuned GPT2 language model (this important detail was mentioned in the appendix). However, currently the graph is over complete and very noisy. The proposed model then sparsifies the graph by learning edge weights via a two-step message passing process. The edge weight is learned as a function of the current edge representation and the textual representation. Lastly, an entropy term is added to the objective function to encourage more peakiness (and sparsity).

The model is tested on three common-sense QA benchmarks and on three of them they beat the baselines albeit only around 1-2%. Statistical significance of the result was not reported. Ablation study show that the efficacy of including the generated edges and pruning the graph. There was a small human-study also done where the annotators were shown a binarized graph and were asked to rate each edge. Annotators had moderate agreement between themselves in finding that the pruned graph was better than the original retrieved graph.


Strengths:
* Developing models that can use symbolic external knowledge present in common-sense KGs and also overcome the sparsity in KG is important and this model is a step in that direction
* The paper achieves a little improvement in performance in all three datasets and ablation experiments are helpful in understanding the results
* The paper is clearly written and it was easy to follow for the most part

Weaknesses & clarifying questions for the authors:
* My biggest complain of the paper in its current form is that several modeling choices were not motivated at all. For example, generating edges between nodes using GPT-2 language model is fairly non-standard. However, the paper lacks any motivation on why this is the right approach to generate facts which are not captured in a KG. What is the guarantee that GPT-2 will not hallucinate and generate a false fact and thereby adding unnecessary noise in the reasoning process.
* Following up on the previous point, there could have been several other modeling choices. For example, instead of generating text via a language model, one could gather text (sentences) from Wikipedia or other text corpora containing the entities (which would mean the text would probably not be a false fact). These modeling choices were not explored and were not discussed.
* The GPT-2 modeling choice was also moved to the appendix and I think it should definitely be moved to the main section of the paper as it is one of the core technical contribution of the paper.
* Another modeling decision that was not motivated was the graph reasoning part. It is unclear to me why the edge weight is modeled as a part of the message passing process. Another (simpler) alternative could be modeling it as an edge attention, which is computed wrt the text and the current node embeddings. I would be curious to know how this simple model worked and if it didn’t why was the case.
* Even though there are improvements across dataset, the improvements are relatively minor (<1% in few datasets). I think it would be useful to have statistical significance test.
* Regarding the human study, if I understand correctly, was only the node and adjacent matrix shown to the annotators?. Was the relation type (KB relations and generated sentences) included too? If they were not included I think they should be because knowing the relations is also very improvement.
* Can you elaborate on the average helpfulness score of edges in table 5? How many (what proportions) were scored 0, 1 or 2 for both the graphs? I think it would also be helpful to report how many facts all/majority of the annotators found to be helpful for both the graphs.

Missing Reference: It would be nice to cite Sun et al EMNLP 2019 -- PullNet: Open Domain Question Answering with
Iterative Retrieval on Knowledge Bases and Text since one of the core contributions of that paper was to retrieve and keep only relevant facts from the KG. Relation paths in KG were explored by several works before Wang et al 2020 such as Neelakantan et al ACL 2015 - Compositional Vector Space Models for Knowledge Base Completion, Das et al EACL 2017 -- Chains of reasoning over entities, relations and text etc. It would be nice to cite those work as well.

Recommendation:  In light of the current weaknesses of the paper, I am giving it a score of 5 and I look forward to the discussion.

=======11/22======

I am deciding to keep the same scores as before. Some of the initial concerns remain. I think the paper still lacks motivation wrt the GPT2 model generating missing edges. Thank you for getting the latest results, the paper is stronger than before and with some more work, I am confident it will be a good contribution to the research community.

=====11/24======

After having read through the explanation behind using GPT2 as edge features (and sufficient backing by 2 closely related work), I am increasing my score to 6. I think the discussion helped in somewhat convincing me that this approach would work for ConceptNet because of its limited schema.

---

> ### Author Response · Authors · 2020-11-20
> **Response to Reviewer 3 (1/3)**
>
> Thank you for your thoughtful review and constructive feedback! We have incorporated your comments to update our paper.
>
> ### Weaknesses & Questions 1
>
> >My biggest complaint of the paper in its current form is that several modeling choices were not motivated at all. For example, generating edges between nodes using GPT-2 language model is fairly non-standard. However, the paper lacks any motivation on why this is the right approach to generate facts which are not captured in a KG. What is the guarantee that GPT-2 will not hallucinate and generate a false fact and thereby adding unnecessary noise in the reasoning process.
>
> Generating facts given a pair of concepts is essentially a KG completion task. Unlike conventional KGs, commonsense KGs are much sparser, which poses challenges for adopting standard embedding-based approaches. As analyzed in Malaviya et al. (2019), an encyclopedic KG like FB15K-237 has 100x the density of a commonsense KG like ConceptNet. Recent works (Malaviya et al., 2019; Bosselut et al., 2019) show that pretrained language models can effectively tackle the sparsity challenge through well-learned semantics of concepts captured during large-scale pretraining. Also, pretrained language models themselves have proven to possess certain commonsense knowledge (Davison et al., 2019). These features make pretrained language models a reasonable choice for commonsense KG completion tasks and motivate our design for the edge feature generator.
>
> As two examples on applying language models for KG completion, COMET (Bosselut et al., 2019) and PathGenerator (Wang et al., 2020) fine-tune GPT and GPT-2 on KG fact triples (or its template-based sentences) to generate novel facts and achieve impressive results. That motivates us to implement the edge feature generator with GPT-2. The major difference between our generator and Wang et al. (2020) is that we use the generator to predict a relation while they use it to predict a path. We admit that noisy facts could be generated by GPT-2. We therefore propose to jointly learn the graph structure to minimize the impact brought by noisy edges generated by the model. In the second example of case study (Appendix §C), the generated facts that are kept after pruning include (office worker, AtLocation, water cooler), (worker, AtLocation, water cooler), (gossip, AtLocation, water cooler). All of them make sense in the context provided by the question-answer pair ("Where would you find an office worker gossiping with their colleagues?", "water cooler").
>
> ### Weaknesses & Questions 2
>
> >Following up on the previous point, there could have been several other modeling choices. For example, instead of generating text via a language model, one could gather text (sentences) from Wikipedia or other text corpora containing the entities (which would mean the text would probably not be a false fact). These modeling choices were not explored and were not discussed.
>
> The characteristics of retrieved textual knowledge are touched in the second paragraph of the Introduction section. A sentence usually contains many concepts and the highly unstructured nature makes it difficult and error-prone to induce the relation between two mentioned concepts. What's more, as commonsense knowledge is usually assumed by humans, most of the commonsense facts are not explicitly written down, especially in Wikipedia which collects encyclopedic knowledge. These issues make retrieved sentences less ideal to be used as features for edges, which are supposed to capture the atomic relational knowledge between concepts. An alternative retrieving source may be OPIEC (Gashteovski et al., 2019), which is a corpus that stores "semi-structured" knowledge in Wikipedia extracted by OpenIE techniques. Given a pair of concepts, we plan to retrieve sentences from OPIEC and encode them as the edge feature. We will report back with results and analysis. Please also feel free to suggest any other experiment settings that you think are more reasonable.
>
> ### Weaknesses & Questions 3
> > The GPT-2 modeling choice was also moved to the appendix and I think it should definitely be moved to the main section of the paper as it is one of the core  technical contributions of the paper.
>
> Thanks for pointing it out! Besides the reason for the space limit, we moved it to appendix because we thought our main model is agnostic to the implementation of the edge feature generator. We agree with your points that it definitely should be put into the main section as it's an important component of the framework and also part of the technical contribution. We have accordingly updated the draft.

---

> > ### Author Response · Authors · 2020-11-20
> > **Response to Reviewer 3 (2/3)**
> >
> > ### Weaknesses & Questions 4
> > > Another modeling decision that was not motivated was the graph reasoning part. It is unclear to me why the edge weight is modeled as a part of the message passing process. Another (simpler) alternative could be modeling it as an edge attention, which is computed wrt the text and the current node embeddings. I would be curious to know how this simple model worked and if it didn't why was the case.
> >
> > We understand the “edge attention” you suggested as an attention mechanism that assigns different scores to different nodes based on node embeddings and the statement vector, and then normalizes the scores in a local neighborhood. It can be implemented as an extension of the graph attention network (GAT) that considers node embeddings and a statement vector. For simplicity, we denote it as "GAT".
> >
> > In contrast, our edge weights design can be understood as a global attention mechanism. The model determines the score for each edge and performs normalization over all edges in the contextualized KG. Besides different designs on how to calculate the unnormalized edge scores, we choose global normalization instead of local normalization because local normalization assumes at least one edge should be helpful in a node's incoming neighborhood, which is not true in our situation. For example, for the distracting or wrongly-grounded concepts (e.g. concept “air” in the example from response to “Weaknesses & Questions 7”), none of their connected edges could be helpful to reasoning and all edges in the neighborhood should be pruned. Those noisy nodes should be softly excluded from message passing (results of global normalization) rather than still receive message from a weighted combination of their neighbors (results of local normalization).
> >
> > We also do experiments to better understand the difference between these two designs. Results suggest that our edge weight design is superior to the local edge attention.
> >
> > | **CommonsenseQA**	| RoBERTa     | | OpenbookQA	| RoBERTa       |
> > |-------------------|-------------| |-------------|---------------|
> > | **GAT**			| 71.20(±0.72)| |    	**GAT**		| 65.10(±0.77)  |
> > | **HGN**			| **72.88(±0.83)**|	| 	**HGN**		| **69.00(±0.95)**  |
> >
> > **New_Table 3. Comparison between our model with a variant using GAT-like local edge attention.**
> >
> > ### Weaknesses & Questions 5
> > >Even though there are improvements across datasets, the improvements are relatively minor (<1% in few datasets). I think it would be useful to have statistical significance tests.
> >
> > Thanks for your suggestion! You can refer to the general response for the discussion on improvements and significant tests. We will also include that in our future version.
> >
> > ### Weaknesses & Questions 6
> > >Regarding the human study, if I understand correctly, was only the node and adjacent matrix shown to the annotators?. Was the relation type (KB relations and generated sentences) included too?
> >
> > Relation types are included in the human study. We presented facts to the annotators for evaluation of validness and helpfulness and a fact takes the form of a triple which describes the relation between two concepts.
> >
> > ### Weaknesses & Questions 7
> > >Can you elaborate on the average helpfulness score of edges in table 5? How many (what proportions) were scored 0, 1 or 2 for both the graphs? I think it would also be helpful to report how many facts all/majority of the annotators found to be helpful for both the graphs.
> >
> > We introduce the criteria for helpfulness score in §4.5 and here we'd like to present the example which we gave to the annotators to make it clearer. For a question-answer pair, ("A man wants air conditioning while we watches the game on Saturday, where will it likely be installed?", "house"), all facts in our learned graph structure include (air conditioning, AtLocation, house), (man, RelatedTo, house), (game, RelatedTo, house), and (air, AtLocation, house). Among them, (air conditioning, AtLocation, house) gets score 2 as this fact corresponds to exactly what the question is asking for -- it "directly leads to the answer". (man, RelatedTo, house) and (game, RelatedTo, house) get score 1 because although they are relevant to the discussed topic, they are not conclusive evidence that justifies the answer. (air, AtLocation, house) gets score 0 as it's irrelevant to the discussed topic, which is caused by redundant recognized concepts.
> >
> > There are 30 graphs in the user study, each graph contains multiple facts, and each fact is rated by 5 annotators. We average all scores (5 annotators * # of facts in 30 graphs) to get the average helpfulness score.
> >
> > For extracted graph structures, on average 26% facts get score 2 (very helpful) and 41% facts get score 1 (moderate helpful). For our learned graph structures, on average 42% facts get score 2 and 36% facts get score 1. This suggests that our model can learn a graph structure with a larger portion of helpful facts.

---

> > > ### Author Response · Authors · 2020-11-20
> > > **Response to Reviewer 3 (3/3)**
> > >
> > > ### Missing Reference
> > >
> > > Thanks for suggesting these papers. In our updated draft, we discuss them when we introduce our methodology (§3.1) and related work (§5). Sun et al. (2019) propose PullNet, which iteratively constructs a question-specific subgraph which contains information relevant to the question. They work on open-domain question answering where retrieved factual knowledge plays a decisive role in inferring the answer entity. However in our task of multiple-choice commonsense question answering, the collected commonsense knowledge is for offering helpful background facts of mentioned concepts to enhance context understanding. Neelakantan et al. (2015) and  Das et al. (2017) leverage the path connecting two entities to infer the relation between them for KG completion. They can be used as an edge feature generator but still face the challenges brought by characteristics of commonsense KGs (as discussed in the response to Weaknesses & Questions 1).
> > >
> > > -----------------------------------
> > > Reference:
> > > - Malaviya et al., AAAI 2020: Exploiting Structural and Semantic Context for Commonsense Knowledge Base Completion
> > > - Davison et al., EMNLP 2019: Commonsense Knowledge Mining from Pretrained Models
> > > - Wang et al., EMNLP 2020: Connecting the Dots: A Knowledgeable Path Generator for Commonsense Question Answering
> > > - Bosselut et al., ACL 2019: COMET: Commonsense Transformers for Automatic Knowledge Graph Construction
> > > - Gashteovski et al., AKBC 2019: OPIEC: An Open Information Extraction Corpus
> > > - Sun et al., EMNLP 2019: PullNet: Open Domain Question Answering with Iterative Retrieval on Knowledge Bases and Text
> > > - Neelakantan et al., ACL 2015: Compositional Vector Space Models for Knowledge Base Completion
> > > - Das et al., EACL 2017: Chains of Reasoning over Entities, Relations, and Text using Recurrent Neural Networks

---

> > > ### Comment · AnonReviewer3 · 2020-11-23
> > > **Response**
> > >
> > > Q4: It is nice that the current method outperforms GAT, however, in the draft of the paper, the modeling decision was not motivated as global reasoning. I think the paper would need more motivation about why a global model is required and why is this the best choice for a global model.
> > >
> > > Q5: Thanks for doing the significance test. If I understand correctly, these tests are done over the results of the latest model (i.e. with PathGenerator)?. Were the results of the original HGN model not significant?
> > >
> > > Q6 and 7: Thanks for the clarification.
> > >
> > > =======11/22======
> > >
> > > I am deciding to keep the same scores as before. Some of the initial concerns remain. I think the paper still lacks motivation wrt the GPT2 model generating missing edges. Thank you for getting the latest results, the paper is stronger than before and with some more work, I am confident it will be a good contribution to the research community.
> > >
> > > =====11/24======
> > >
> > > After having read through the explanation behind using GPT2 as edge features (and sufficient backing by 2 closely related work), I am increasing my score to 6. I think the discussion helped in somewhat convincing me that this approach would work for ConceptNet because of its limited schema.

---

> > > > ### Author Response · Authors · 2020-11-24
> > > > **Response to Q4, Q5**
> > > >
> > > > >Q4: It is nice that the current method outperforms GAT, however, in the draft of the paper, the modeling decision was not motivated as global reasoning. I think the paper would need more motivation about why a global model is required and why is this the best choice for a global model.
> > > >
> > > > Thanks for your suggestions! We have added detailed discussion about motivations for global edge weight and local edge attention in §3.2 “Reasoning with Weighted Graph” as well as experiment results in §4.3 “Study on More Model Variants” in the updated draft. We also updated our response to “Weaknesses & Questions 4” to make it clearer.
> > > >
> > > > >Q5: Thanks for doing the significance test. If I understand correctly, these tests are done over the results of the latest model (i.e. with PathGenerator)?. Were the results of the original HGN model not significant?
> > > >
> > > > The significance test in the general response is done on HGN (w PathGenerator).
> > > >
> > > > For our original HGN:
> > > > - on 60% CommonsenseQA + BERT-Base: significant over all baselines
> > > > - on 100% CommonsenseQA + BERT-Base: significant over all baselines
> > > > - on 60% CommonsenseQA + BERT-Large: significant over all baselines except MHGRN
> > > > - on 100% CommonsenseQA + BERT-Large: significant over all baselines except MHGRN
> > > > - on 60% CommonsenseQA + RoBERTa: significant over all baselines except MHGRN
> > > > - on 100% CommonsenseQA + RoBERTa: significant over all baselines except PathGenerator
> > > > - on OpenbookQA + RoBERTa: significant over all baselines except PathGenerator
> > > > - on OpenbookQA + AristoRoBERTa: significant over RN, RGCN, GconAttn
> > > > - on CODAH: significant over RGCN on RoBERTa
> > > >
> > > > Note that MHGRN and PathGen are both to be published in EMNLP 2020. For CODAH, we analyze in §4.3 “Performance Comparisons” that: “As Chen et al. (2019) suggest, questions in CODAH mainly target commonsense reasoning about quantitative, negation and object reference. In this case, relational knowledge provided by ConceptNet may only offer limited help.”
> > > >
> > > > We do observe more significant improvement when equipping our model with PathGenerator (without hyperparameter tuning) as presented in the general response. Other advantages and values of our model beyond performance gain are also discussed there. We’ll do more exploration and tuning on our HGN (w PathGenrator) model to include stronger results.
> > > >
> > > > > Some of the initial concerns remain. I think the paper still lacks motivation wrt the GPT2 model generating missing edges.
> > > >
> > > > Thanks for bringing this up! We have included more details and motivations for GPT-2 based edge feature generator in §3.1 “Edge Feature Generator” based on our discussion. We hope our updated draft could address your concern. If there’s still something unclear, we appreciate it if you could provide more feedback.

---

> > ### Comment · AnonReviewer3 · 2020-11-23
> > **Response**
> >
> > Thank you for the detailed response to my review.
> >
> > Q1: An entity pair can have multiple relations between them. For example, two entities can be spouse, and friends, and co-workers. It is unclear to me, just giving [h and t] as a prompt to GPT2, would generate all possible relations between them. I think, ideally, you would need to give more context than the name of the entities. I am also still not clear how learning graph structure jointly will help mitigate cases when GPT2 would generate wrong facts.
> >
> > Q2: I agree with you regarding the reporting bias wrt common-sense concepts in text. However, the LMs that you are using to generate facts have only been trained on a lot of text, so I am not sure, how generating text from pre-trained LMs helps you overcome that. The experiment you suggested makes sense. I would also try to retrieve text containing entities conditioned on the context and generate text features (using LMs) rather than generating text from LMs and featurizing them.

---

> > > ### Author Response · Authors · 2020-11-24
> > > **Response to Q1**
> > >
> > > >Q1: An entity pair can have multiple relations between them. For example, two entities can be spouse, and friends, and co-workers. It is unclear to me, just giving [h and t] as a prompt to GPT2, would generate all possible relations between them. I think, ideally, you would need to give more context than the name of the entities. I am also still not clear how learning graph structure jointly will help mitigate cases when GPT2 would generate wrong facts.
> > >
> > > Thanks for your question. We want to highlight that the characteristics of ConceptNet are very different from conventional KGs like DBPedia. For nodes, while DBPedia focuses on named entities, ConceptNet “puts more emphasis on collecting information about common words than about named entities” (Speer and Havasi, 2013). For edges, while DBPedia contains thousands of relations, ConceptNet only defines 34 **basic** relations which are listed in https://github.com/commonsense/conceptnet5/wiki/Relations. These predefined basic relations are mostly exclusive. In ConceptNet, 95.3% of all concept pairs only have one relation between them, which means we can approximately define the relation prediction (KG completion) task as a single-label classification task. Therefore, even if there could be multiple relations between an entity pair, most of them will not be captured in ConceptNet. Relations like “spouse”, “friend”, “co-worker” are too specific. They are captured by DBPedia but not by ConceptNet. We choose ConceptNet as our knowledge source because we want to incorporate commonsense knowledge about concepts (e.g. (book, AtLocation, house)) rather than world knowledge about named entities (e.g. (Apple, /business/company/founders, Steve Jobs)).
> > >
> > > The role of the edge feature generator is to do the KG completion task where we want to predict **one relation out of 34 predefined ConceptNet relations** given a pair of concepts. In other words, given [h, t] as a prompt, the GPT2 model is only asked to generate one sentence in the form of [h, r, t] that describes the possible ConceptNet relation between them. This is like doing on-the-fly KG completion to incorporate missing edges when we construct the contextualized KG. Note that although r is predicted in a generative way, it will still be one of 34 ConceptNet relations, because **GPT2 is finetuned on ConceptNet facts** and can be easily fitted to only predict ConceptNet relations. This module is not designed to be context-aware for two reasons:
> > >
> > > - Commonsense knowledge itself is independent of any context. For example, (book, AtLocation, house) is a commonsense fact just because it holds true in most cases in people’s lives. Though being uncontextualized, it “lays a critical foundation without which more nuanced interpretation cannot exist” (Liu and Singh, 2004). What should be contextualized is the incorporation of certain commonsense facts --- that’s why we do pruning after collecting the facts by determining whether they are helpful for reasoning.
> > >
> > > - It’s hard to get training data that has both the commonsense fact and the context that needs this fact.
> > >
> > > GPT2 is a plug-in module. We don’t change it after it has been finetuned on ConceptNet facts. We just use it to generate facts between concept pairs of interests. The generated facts correspond to edges in the contextualized KG. They can be clean or noisy, helpful or unhelpful. We jointly learn the edge weights (graph structure) with the reasoning parameters, so that the edges will be reweighted according to their helpfulness for reasoning. Figure 6 in the paper gives an example. We should have a contextualized KG with fully-connected edges between question and answer concepts after graph initialization (§3.1). By using our joint learning approach, wrong facts and unhelpful facts (41 out of all 48 extracted and generated facts) are softly removed.
> > >
> > > ---
> > > Reference:
> > > - Speer and Havasi, 2013: ConceptNet 5: A Large Semantic Network for Relational Knowledge
> > > - Liu and Singh, 2004: ConceptNet—a practical commonsense reasoning tool-kit

---

> > > ### Author Response · Authors · 2020-11-24
> > > **Response to Q2**
> > >
> > > >Q2: I agree with you regarding the reporting bias wrt common-sense concepts in text. However, the LMs that you are using to generate facts have only been trained on a lot of text, so I am not sure, how generating text from pre-trained LMs helps you overcome that. The experiment you suggested makes sense. I would also try to retrieve text containing entities conditioned on the context and generate text features (using LMs) rather than generating text from LMs and featurizing them.
> > >
> > > Sorry for the confusion. The GPT2 model is pretrained on large corpora and then **finetuned on ConceptNet facts in the prompt-generation format** (described in §C in the original version). The finetuning process makes it no longer a general-purpose language model. The finetuned GPT2 can generalize ConceptNet facts to more novel commonsense facts because it can leverage the rich semantic and relational knowledge learned during pretraining and fineuning. As an intuitive example, a pretrained language model will assign a higher probability to “book is located in house” (corresponding to the correct fact (book, AtLocation, house)) than “book is a house” (corresponding to the wrong fact (book, IsA, house)). The knowledge it captures during finetuning is very helpful for the commonsense KG completion task.
> > >
> > > For the experiments on retrieving facts from OPIEC instead of generating facts using GPT2, we name it as “HGN (w OPIEC)”. Given a pair of concepts that are non-adjacent in ConceptNet, we want to retrieve all triples describing their relations from OPIEC based on string matching. If there's no such triple, then these two concepts won't be connected in the contextualized knowledge graph. If there's at least one matching triple, then we convert each triple into a natural language sentence and adopt a BERT-base model (which has the same hidden dimension as GPT2 that enables a fair comparison) to encode the concatenation of all the sentences, which will be used as the edge feature. The results are shown in New_Table 4, suggesting that generated facts are more helpful than retrieved facts. Generated facts are more likely to fill the knowledge gap for reasoning than retrieved facts because they don’t have the coverage issue as the retrieved facts.
> > >
> > > | **CommonsenseQA**	| RoBERTa     | | OpenbookQA| RoBERTa       |
> > > |:-------------------|:-------------:|-|:-------------|:---------------:|
> > > | **HGN (w OPIEC)**	| 71.35(±0.21)| |    	**HGN (w OPIEC)**	| 67.95(±1.23) |
> > > | **HGN**		| **72.88(±0.83)**|| 	**HGN**	| **69.00(±0.95)**  |
> > >
> > > **New_Table 4. Comparison between our model with a variant using retrieved facts from OPIEC.**
> > >
> > > Thank you for suggesting the context-aware retrieval approach. We agree that it’s a great direction to explore. While this approach is trying to get more contextualized facts at the graph initialization stage, our work is more focused on how to jointly denoise the collected facts at the graph reasoning stage. As it’s non-trivial to retrieve contextualized **atomic relational knowledge** given two concepts and their context, we will leave it as future work.

---

### Official Review · AnonReviewer2 · 2020-10-30
**Novelty of the proposed model is limited**

**Rating:** 5
**Confidence:** 3

**Review:**

The paper proposes a graph network (called HGN), aiming to better leverage commonsense knowledge graphs (KGs) to solve commonsense question answering and reasoning tasks, by jointly generating representations for new triples from KGs, determining relevance of the triples, and learning graph model parameters. The proposed model is tested on several tasks: CommonsenseQA, OpenbookQA, and CODAH.

Pros:
-  Overall, the paper is easy to follow, although there are a number of typos or grammatical errors that need to be fixed.  The overall idea is clear.
-  Jointly learning (pruning) the graph structure with the network parameters is interesting.
-  The proposed model outperforms the baselines in comparison.
-  Human evaluation is provided.

Cons:
-  My major concern about this paper is the novelty and contributions in terms of methodology. Compared to existing methods (e.g., those PG models proposed (Wang et al. 2020)), the novelty of the current submission is rather limited---the proposed model of jointly generating new triples and learning (pruning) the graph structure with the network parameters is an interesting, but a pretty incremental idea.

-  The empirical comparison to previous work (e.g., Wang et al. 2020) needs to be clearer to help understand the empirical advantages of the proposed models. The paper mentioned some reason of excluding PG-Full from comparison, but since PG-Global does not include static knowledge embedding and PG-full does, is the latter a more reasonable baseline to be compared with? The model does not achieve better performance than existing models on some tasks, which casts doubts on its effectiveness; e.g., whether its advantage is orthogonal to that brought by stronger models such as those performing much better on the OpenbookQA task.

More comments:
-  The paper uses much space to discuss neural symbolic approaches. Given the vague benefit of doing so, it may be better to use the limited space to focus more on establishing the contributions w.r.t. existing models; e.g., more details about (Wang et al., 2020) can be provided and compared to in both methodological and experimental analyses.
- The human evaluation was performed on the questions with correct questions. More analyses on the edges and weights generated for questions that were not correctly answers may help better understand the proposed model.

---

> ### Author Response · Authors · 2020-11-20
> **Response to Reviewer 2 (1/3)**
>
> Thank you for your thoughtful comments!
>
> ### Cons 1
>
> > My major concern about this paper is the novelty and contributions in terms of methodology. Compared to existing methods (e.g., those PG models proposed (Wang et al. 2020)), the novelty of the current submission is rather limited --- the proposed model of jointly generating new triples and learning (pruning) the graph structure with the network parameters is an interesting, but a pretty incremental idea.
>
> Thank you for the question regarding the novelty of our work. We have included more discussions about the novelty in the updated draft (mainly in §1) and also recap our contributions here.
>
> The quality of the collected evidence facts plays a vital role in KG-augmented commonsense reasoning but has been overlooked by previous works. Enriching KG facts with generated facts and pruning unreliable and unrelated facts is an important problem for maintaining strong reasoning performance. However, how to jointly manipulate (prune) the graph structure while performing message passing-based reasoning in a graph network (along with edge features) is a non-trivial problem --- there’s no direct supervision guiding us to either keep or remove certain facts and existing works simply assume a *static graph* is used throughout the learning process. We therefore propose to jointly learn the graph structure using the downstream task as the signal. To our knowledge, our work is one of the first to jointly conduct graph structure pruning and parameter learning on graph networks (along with updating edge features) for KG-augmented commonsense reasoning. Our comparisons to baseline models show that our joint learning approach helps obtain stronger performance.
>
> Compared with path-based methods including PathGenerator, we also want to clarify our novelty claim is not on the “triple generation” module alone, but more on jointly learning the graph structure with the parameters for reasoning (contributions summarized in the last paragraph of §1). While previous works study how to reason over an extracted graph and Wang et al. (2020) focus on how to generate new paths as evidence, they all reason over a static graph, which is assumed to have a “clean” structure. Our work is the first to drop this problematic assumption, and integrate both extracted and generated facts into a unified graph reasoning model with the denoising ability.
>
> To summarize, our work resolves three intrinsic issues with KG-augmented commonsense reasoning models:
>
> 1.  low coverage of KG facts: we generate facts to complete the contextualized KG.
>
> 2. limited expressiveness of KG relations: we generate continuous relational features instead of embedding the generated relations with a lookup table for better expressiveness.
>
> 3. wrong facts or uncontextualized facts (facts that contain mentioned concepts but are not helpful for reasoning): we jointly prune the graph structure by edge reweighting with entropy regularization during reasoning.
>
> We consider 3 as our major novelty and uniqueness from previous and contemporaneous works --- we no longer assume the perfect graph structure of a contextualized KG, which is too good to be true and hinders the model from benefiting from high-quality supporting facts.

---

> > ### Author Response · Authors · 2020-11-20
> > **Response to Reviewer 2 (2/3)**
> >
> > ### Cons 2
> >
> > >The empirical comparison to previous work (e.g., Wang et al. 2020) needs to be clearer to help understand the empirical advantages of the proposed models. The paper mentioned some reason of excluding PG-Full from comparison, but since PG-Global does not include static knowledge embedding and PG-full does, is the latter a more reasonable baseline to be compared with? The model does not achieve better performance than existing models on some tasks, which casts doubts on its effectiveness; e.g., whether its advantage is orthogonal to that brought by stronger models such as those performing much better on the OpenbookQA task.
> >
> > Thanks for your question! The PG-Full model is described as:
> > >(3) PathGeneratorFull (or PG-Full): We equip our reasoning module with both the global path generator and the RN baseline described above. In specific, we extend Eq. 8 by feeding the concatenation of the context embedding, the path embedding and the static knowledge embedding to the classifier." (https://arxiv.org/abs/2005.00691v1)
> >
> > The authors concatenate the knowledge embedding generated by PG-Global model ("the path embedding") and the knowledge embedding generated by RN model ("the static knowledge embedding") as the final knowledge embedding used for making predictions. Since it directly combines the features generated by PG-Global and RN, we consider it as an ensemble model of these two models, instead of a unified graph reasoning model that operates on both generated and extracted facts. We think it's unfair to compare our single HGN model with their ensemble PG-Full model.
> >
> > To better support our argument, we build a variant of our model named "HGN+RN" by combining the features generated by our model and RN, with the statement vector ("context embedding"). New_Table 2 shows that we can also get improvement by integrating RN. HGN+RN significantly outperforms PG-Full with p-value < 0.05. That makes us think that it's more meaningful to do comparison among "single" models by excluding the "ensemble" one.
> >
> > | CommonsenseQA		| RoBERTa| | CommonsenseQA		| RoBERTa|
> > |:--------------------------|:---------------:|--|:--------------------------|:---------------:|
> > | **PathGenerator (PG-Global)** | 71.55(±0.99)| | **PG-Full (PG-Global+RN)**    | 72.68(±0.42)|
> > | **HGN**                       | **72.88(±0.83)**| | **HGN+RN**                    | **73.43(±0.26)**|
> >
> > **New_Table 2. Effect of adding features generated by RN to our model.**
> >
> > Regarding the effectiveness of our proposed model on OpenbookQA, we observe consistent improvement over all baselines (with the contemporaneous work Wang et al., (2020) as the only exception) when we shift our text encoder from RoBERTa to AristoRoBERTa, which implies our advantage is orthogonal to that brought by stronger base models. As shown in the general response, our HGN outperforms PathGenerator with the AristoRoBERTa encoder when equipped with a stronger edge feature generator. On OpenbookQA's leaderboard, our original HGN achieves the highest test accuracy (81.4) among all models using AristoRoBERTa as the text encoder, which is higher than PathGenerator (80.4). Submissions using different text encoders are not comparable. We choose AristoRoBERTa instead of AristoAlbert as the text encoder because:
> > 1. AristoRoBERTa is a well-established model and used by many other competitors so we can have fair comparisons with more models;
> > 2. the training of AristoAlbert is about 8 times slower than AristoRoBERTa, making it less practical to be adopted.
> >
> > More discussion on the empirical effectiveness of our model is presented in the general response, where we adopt a stronger edge feature generator and achieve significant performance gains.

---

> > > ### Author Response · Authors · 2020-11-20
> > > **Response to Reviewer 2 (3/3)**
> > >
> > > ### Comments 1
> > >
> > > >The paper uses much space to discuss neural symbolic approaches. Given the vague benefit of doing so, it may be better to use the limited space to focus more on establishing the contributions w.r.t. existing models; e.g., more details about (Wang et al., 2020) can be provided and compared to in both methodological and experimental analyses.
> > >
> > > Thanks for the suggestion! In the updated draft, we have compressed the discussion and made §2 a more compact background section that presents the problem formulation, notations, and high-level architecture. We discuss the design difference when we introduce our methodology (the last paragraph of §2; §3.1 "Edge Feature Generator"). We highlight Wang et al. (2020) by putting it into a category named "Models Using Generated Facts" and providing more details when we introduce the compared methods (§4.2). We also make it clear in §5 that the core difference between our work and Wang et al. (2020) is that they still assume a static graph structure for reasoning. We will formally add more empirical discussion in the future version where we also adopt the path generator (Wang et al., 2020) as an implementation of our relational feature generator f_{gen}.
> > >
> > > ### Comments 2
> > >
> > > >The human evaluation was performed on the questions with correct questions. More analyses on the edges and weights generated for questions that were not correctly answered may help better understand the proposed model.
> > >
> > > Sorry for the confusion. We don't distinguish correctly-answered questions from wrongly-answered questions in the user study --- we looked at both of them. Specifically, in CommonsenseQA, each question has five answer candidates and these five (question, answer candidate) pairs will give five graphs. Among these five graphs, we want to select the one that corresponds to the question with the correct answer candidate. This is because the contextualized KG provides evidence facts in a supportive way --- plausible answer candidates get more supportive facts while implausible answer candidates get fewer. Therefore, analyzing how a correct answer candidate can be better supported by our hybrid facts compared to extracted facts could be a more intuitive way to understand the effectiveness of our graph structure learning strategies. On the contrary, wrong answer candidates are usually less relevant to the question concepts. The contextualized KG is much sparser with less room to be improved with supportive facts.
> > >
> > > To better address your concern, we also investigate the questions that are wrongly answered by our model. We identify two patterns from these wrongly answered questions:
> > >
> > > \* edge format: (subject, relation, object, edge weight)
> > > 1. The wrong answer is associated with many more facts than the correct answer.
> > >
> > > 	*Example:*
> > > 	- Question: What would encourage someone to continue playing tennis?
> > > 	- Wrong answer (model's prediction): exercise
> > > 		- Edges (weight ≥ 0.01): (play tennis, Causes, exercise, 0.47), (playing tennis, Causes, exercise, 0.47), (tennis, IsA, exercise, 0.04), (play, RelatedTo, exercise, 0.01), (playing, UsedFor, exercise, 0.01)
> > > 	- Correct answer: victory
> > > 		- Edges (weight ≥ 0.01): (playing tennis, Causes, victory, 0.99)
> > > 	- Analysis: Answers with more supporting facts are more likely to be the correct one in the training set. That may make the model biased towards choosing the answer associated with many more facts regardless of their relevance to the question. Some kind of normalization could be helpful to alleviate this issue.
> > >
> > > 2. The crucial fact leading to the correct answer is not identified and upweighted.
> > >
> > > 	*Example:*
> > > 	- Question: A revolving door is convenient for two direction travel, but it also serves as a security measure at a what?
> > > 	- Wrong answer (model's prediction): mall
> > > 		- Edges (weight ≥ 0.01): (revolving door, AtLocation, mall, 0.98)
> > > 	- Correct answer: bank
> > > 		- Edges (weight ≥ 0.01): (revolving door, AtLocation, bank, 0.94), (security, RelatedTo, bank, 0.03), (two, RelatedTo, bank, 0.03)
> > > 	- Analysis: Both the correct answer and the wrong answer has a revolving door, making (security, RelatedTo, bank) the most discriminative fact. However, the model fails to identify this crucial fact and upweight it. This is a weakness brought by separately modeling the question with different answers, which makes it difficult for the model to identify the discriminative facts by comparing different answer choices. A future direction could be modeling all choices within one contextualized KG to make them aware of each other and identify discriminative facts.
> > >
> > > -----------------------------------
> > > Reference:
> > > - Wang et al., EMNLP 2020: Connecting the Dots: A Knowledgeable Path Generator for Commonsense Question Answering

---

> ### Author Response · Authors · 2020-11-24
> **Look Forward to Hearing from You**
>
> Dear Reviewer 2,
>
> Thank you very much for your feedback! In our response, we have included more discussion and new experiments to address your concern. We have also updated our draft based on your suggestion. We're wondering if you have further questions or comments regarding this work? We really appreciate it if you could let us know before the end of the rebuttal period.
>
> Thanks again for your time!

---

### Author Response · Authors · 2020-11-20
**General Response**

We would like to thank all reviewers very much for your valuable feedback and constructive comments. In our updated draft, we have included additional experiments to address reviewers' questions and improve the writing based on the suggestions.

### Summary of Updates

Below we briefly summarize our major updates:

- We replace the edge features in our model with ones generated by a contemporaneous work, PathGenerator (Wang et al., 2020). We compare this variant with baseline models and conduct statistical significance tests. [New_Table 1 in General Response]
- As a fair comparison to PathGenerator-Full (PG-Full), we augment our model with the graph vector generated by RN and compare it with PG-Full. [New_Table 2 in Response to Reviewer 2]
- We compare our method with Graph Attention Network (GAT) to study the design of edge attention. [New_Table 3 in Response to Reviewer 3]
- We compare our method with a variant that encodes retrieved facts as edge features. [New_Table 4 in Response to Reviewer 3]
- We update the draft for fixing typos and grammatical errors and incorporating suggestions for better presentation (especially on motivations and details for the GPT-2 based edge feature generator and edge weights, discussions on PathGenerator and missing reference).

### Performance Gains over Strong Baselines

There are concerns about the performance gains of our method, especially compared to a contemporaneous work, PathGenerator (Wang et al., 2020). The PathGenerator paper proposes a multi-hop path generator (a fine-tuned GPT-2 model) to generate the path between a pair of concepts. We admit that their generator is stronger than the one we use as it captures *multi-hop* relations between concepts. However, we want to highlight that our proposed method can cope with different designs of the edge feature generator (please see footnote 2 in the paper) and can incorporate stronger edge features like the ones proposed by PathGenerator. Our main contribution is on jointly generating edge features, learning the graph structure, and performing reasoning over the pruned graph structure --- we leave the development of more sophisticated edge feature generation as future work.

We want to note that PathGenerator (Wang et al., 2020) is a contemporaneous work that was accepted to EMNLP 2020 on Sep 14 and released code on Oct 3. As suggested by the conference organizers (https://iclr.cc/Conferences/2021/ReviewerGuide), "If a paper was published on or after Aug 2, 2020, authors are not required to compare their own work to that paper. Authors are encouraged to cite and discuss all relevant papers, but they may be excused for not knowing about papers not published in peer-reviewed conference proceedings or journals." We didn't reproduce and adopt their generator given the limited time before the ICLR deadline.

As Wang et al. (2020) have recently released their code, we build our HGN on top of their proposed PathGenerator with the strongest text encoders on two datasets (RoBERTa for CommonsenseQA, and AristoRoBERTa for OpenbookQA).  We summarize the comparison results below, where we observe consistent improvements over all baselines.

|CommonsenseQA|RoBERTa| |OpenbookQA|AristoRoBERTa|
|:---------------------|:------------:| |:---------------------|:-------------:|
| **RN**| 70.08(±0.21)| |**RN**| 75.35(±1.39)|
| **MHGRN**| 71.11(±0.81)| |**MHGRN** | 77.75(±0.38)|
| **PathGenerator**| 71.55(±0.99)| | **PathGenerator**| 80.05(±0.68)|
| **HGN**| 72.88(±0.83)| | **HGN**| 79.00(±1.43) |
| **HGN (w PathGenerator)**| **73.53(±0.67)**| |**HGN (w PathGenerator)**|**80.10(±1.03)**|

**New_Table 1. Performance comparison between our model using PathGenerator as the edge feature generator with strong baseline models on CommonsenseQA and OpenbookQA datasets.**

### Significant Tests on Results of HGN (w PathGenerator) vs Strong Baselines

To address the concerns on the statistical significance of the performance differences (Reviewers 1, 3), we perform unpaired two-tailed t-Tests (with p-value < 0.05). On CommonsenseQA, HGN (with PathGenerator) is significantly better than all baselines. On OpenbookQA, HGN (with PathGenerator) is significantly better than all baselines except PathGenerator.

In addition to outperforming baseline methods, we also want to highlight that our model demonstrates its strong performance under *low-resource settings* (where we observe relatively more performance gains, as shown in Figure 6 in the paper). Besides, our way of explicitly generating, denoising, and reasoning with supporting evidence provides good *interpretability* towards building more trustworthy reasoning systems. Our work also points out a direction towards improving KG-augmented QA framework with a focus on the *coverage and quality of knowledge facts*.

-----------------------------------
Reference:
- Wang et al., EMNLP 2020: Connecting the Dots: A Knowledgeable Path Generator for Commonsense Question Answering

---

### Decision · Program_Chairs · 2021-01-07
**Final Decision**

**Decision:**

Reject

**Comment:**

The paper proposes an interesting step in the direction of neuro-symbolic reasoning. While there is no consensus among reviewers about the key novelty of the method, all acknowledge the interest of the direction. All of them also recognize that the submission improved greatly during the discussion phase: clarification of motivations, of experimental settings and results, of discussion with previous work.

However, despite those improvements, the submission is not yet ready for publication at ICLR. We encourage the authors to use the very detailed reviews and comments to improve the work. In particular, we encourage them to pay attention at three aspects:

1/ Comparison with large language models: the discussion wrt T5 is important. A key motivation for the proposed model is that it is bringing information and elements for QA (or other reasoning tasks) that purely scaling up language models can not bring. Or maybe they can bring the same kind of improvement but at a much lower computational cost. In any case, this is a very important point to justify the interest of such approach, and neuro-symbolic reasoning overall, empirically.

2/ Using GPT2 (or equivalent): the discussion on using GPT-2 for generating new facts is key too. It is essential to bring this description from appendix to the core of the paper. But more discussion are expected.  For instance, what if GPT-2 generates facts that are false and lead to answering and justifying a wrong answer? In other words, how does it impact the integrity of the contextualized KG? This is an essential point that needs to be worked on more thoroughly.

3/ Overall there have been a lot of discussion to improve the motivations and the contributions. But they are not reflected in the paper necessarily. Following R2, we encourage the authors to "refocus the existing version (e.g., from vague discussion about neural-symbolic models towards establishing solid comparison to the most related previous work in various sections of the submission)"